rsos.royalsocietypublishing.org

environmental science/biochemistry/
environmental engineering

ecological simulation, submerged macrophytes,
*Vallisneria*, integrated biomarker response,
Lake Poyang

**Author for correspondence:**
Yong Ji
e-mail: jiyong@nit.edu.cn; 472388451@qq.com

# Integrated biomarker responses of the submerged macrophyte *Vallisneria spiralis* via hydrological processes from Lake Poyang, China

Yong Ji[1], Zhidong Yao[1], Jie Zhang[1,2], Xueru Wang[1], Jixiang Luo[1], Liying Xiao[1,2] and Shifeng Zhang[3]

[1]College of Water Conservancy and Ecological Engineering, Nanchang Institute of Technology, Nanchang 330099, People's Republic of China
[2]College of Environment, Hohai University, Nanjing 210098, People's Republic of China
[3]MOE Key Laboratory of Wooden Material Science and Application, Beijing Forestry University, Beijing 100083, People's Republic of China

YJ, 0000-0002-4604-4258

*Vallisneria spiralis*, a widely distributed wetland plant, was used to reveal how the light intensity at the top of the plant, plant morphology and antioxidant enzyme activity respond to different hydrologic conditions from Lake Poyang, China. By designing a laboratory experiment simulating historical water levels of low, normal and high wetland plant submersion, this study aimed to elucidate the effects of different levels of flooding on growth and antioxidant enzyme activity for *V. spiralis*. The results showed that the plant crown light intensity of the treated group and control group (CG) first decreased and then increased along with the seasonal variation of the water level. The maximum and minimum values of the plant crown light intensity were observed in April and July, respectively. Similar to the CG, *V. spiralis* from the normal and low water level (LWL) groups was measured and had higher plant height growth in the flooding period from May to June, and the entire plant biomass also showed a steady growth trend in the same period. However, the plant growth of the high water level (HWL) group was lower during the whole simulation period, with negative growth in July. Antioxidant enzyme activities changed with the seasonal temperature, and the activity of the CG showed a rising trend. Compared with those of the CG, the antioxidant enzyme activities of the HWL group showed a 'bell shaped' trend, which was first significantly induced and then significantly inhibited.

In addition, the peroxidase (POD) and catalase (CAT) activities from the LWL group in April were also significantly induced. The integrated biomarker response (IBR) index showed that a comprehensive biological index could well reflect the effects of seasonal water levels in Poyang Lake on the growth of the wetland plant *V. spiralis*. This study indicated that high flooding levels had the strongest negative effect on the growth and enzyme activity of the submerged plant *V. spiralis*.

## 1. Introduction

As an important global ecosystem, wetland ecosystems play crucial roles in climate regulation, water conservation, pollution purification, and biodiversity and habitat protection [1]. Among numerous factors affecting the wetland ecosystem, hydrological processes play dominant roles in the formation of the vegetation community and ecological processes, and are the decisive factor in the structure and function of wetland ecosystems [2]. Over the past several decades, the make-up of the wetland environment and vegetation distribution pattern have been profoundly affected and modified by tensely anthropogenic activities and climate change [2]. In particular, the effect of water level fluctuations on the growth of wetland plants and their adaption mechanisms are of growing interest due to high frequency and density effect of hydrological regimes coming from human actions [3,4]. The hydrologic conditions of the wetland water level are an important feature and directly affect the spatial distribution of vegetation, species and their seasonal variation in wetlands [5,6]. The water level can affect the light intensity, which is the energy source of photosynthesis and affects the vegetation photosynthetic rate and, consequently, determines the vegetation height and biomass growth [7,8]. Accordingly, the vegetation community can also respond to hydrological changes by the allocation of biomass aboveground and underground in different hydrological seasons [9].

Under the adapted range of the water level, wetland vegetation can maintain normal growth using an adaptation mechanism. In addition to the growth of morphological responses of vegetation, vegetation can also maintain normal physiological function and growth through a series of complex changes in physiological metabolism [10]. To adapt the negative effects of high water stress and eliminate the free radicals under anaerobic active oxygen, the vegetation has formed a set of complex antioxidant systems in the evolutionary process. Among those antioxidant systems, antioxidant enzyme activities of vegetation are essential to maintain normal growth and development under the stress from high water levels [11]. The three most effective antioxidant enzymes are the superoxide dismutase (SOD), peroxidase (POD), catalase (CAT) and lipid degradation product (thiobarbituric acid, TBARS), which have been studied for a long time and widely used for field assessments [12,13]. However, the antioxidant enzyme activities of vegetation can change dramatically due to the impact of different plants and diverse environmental effects. In some cases, a completely opposite result can even be observed for the same plant in different environments, and this limits the use of antioxidant enzyme activities for pollutant evaluation. Beliaeff & Burgeot [14] mentioned a method that offered an integrated biomarkers comprehensive index (IBR) by combining all indices together, and it has been proved an effective method to evaluate the growth response under exogenous stress [15,16].

As shown in figure 1, Poyang Lake has an area of approximately 3000 $km^2$ at normal water level (NWL) and plays a crucial role in maintaining a dynamic wetland system. As one of the most important wetland habitats for migratory birds in the world, Poyang Lake is rich in wetland vegetation diversity that is directly affected by seasonal water level fluctuations and shows a significant spatial distribution pattern [5,6,17]. In recent years, the Poyang Lake wetland ecological system has been continuously modified under human disturbance with continuously sustained high or low water level (LWL) operations [18]. The original hydrologic conditions of Poyang Lake wetland are quietly changing, which has been observed in recent years after the construction and operation of the Three Gorges Reservoir (TGR) [19]. Protection of the Poyang Lake wetland is becoming more and more important under the potential Ecological Water Conservancy Hydro-Junction project in Poyang Lake. Several papers about the impact of water level fluctuations on wetland vegetation community from Poyang Lake have been published, and the results have definitely indicated that most marshlands experienced an inflection point after the operation of TGR [5,20].

Seasonal and multi-year water level fluctuations in Poyang Lake have been benefited to maintain wetland vegetation diversity. However, little is known about the relationship between water level fluctuations and their effect on the vegetation community and biochemical indices of submerged plants of Lake Poyang. *Vallisneria spiralis* is a perennial submerged herb and one of the dominant

rsos.royalsocietypublishing.org    R. Soc. open sci. **5**: 180729

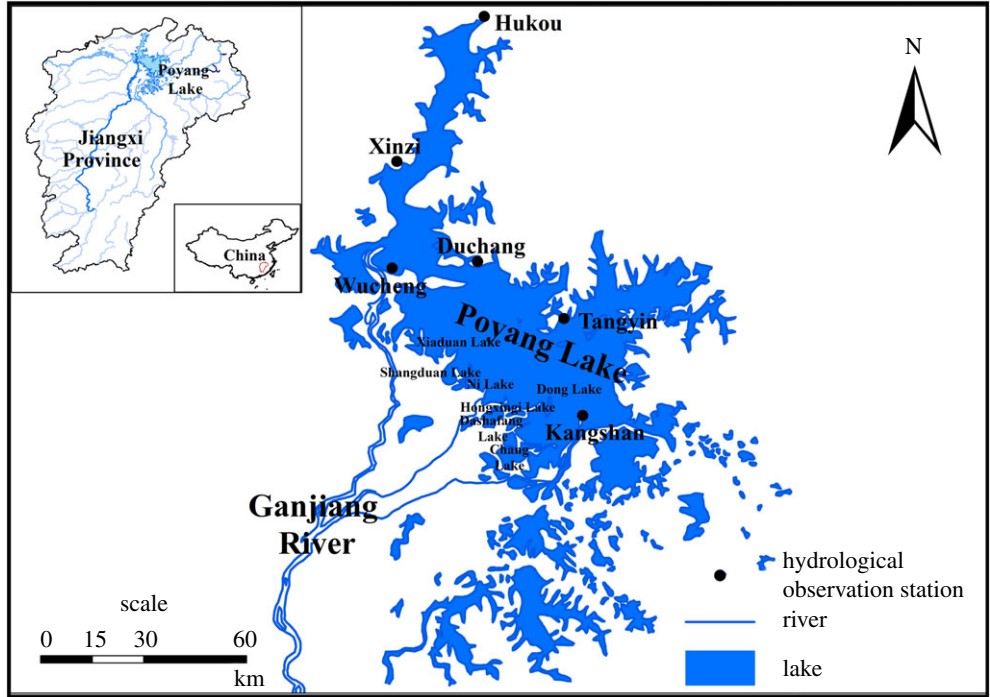

**Figure 1.** Location of Lake Poyang and distribution of Disc Lake, China.

species of common submerged plants in the middle and lower Yangtze River freshwater lakes and wetlands [6]. Owing to its higher nutritional value, *V. spiralis* plays an important role in the ecological system of wetland and bird protection area ecosystems [21]. *Vallisneria spiralis* is also widely distributed in the lower part of Poyang Lake, named the 'Disc Lake' based on its shape, where it is usually separated from the main lake area and forms an individual internal lake in the dry season [6]. The long-term and large-scale human behaviour represented by the TGR has led to significant changes in the submerged vegetation community in Poyang Lake [18–20].

To cope with these negative effects, we hypothesize that the submerged vegetation represented by *V. spiralis* could adapt to the rhythm of wetland hydrological processes by regulation of vegetation morphology, redistribution of nutrients between aboveground and underground parts, and molecular enzyme response mechanisms. To fully elucidate the effects of different hydrological processes on the growth of *V. spiralis* in Lake Poyang, an experiment was designed to simulate three typical hydrological conditions (high level, normal level and low level) based on hydrological data from Poyang Lake. Physical indicators such as light intensity, plant height and biomass allocation, and biological indicators including SOD, POD and CAT were measured during the entire experiment process. Integrated biomarker responses were also calculated to clarify the physiological processes of eelgrass under the different hydrological wetland factors in Poyang Lake.

# 2. Material and methods

## 2.1. *Vallisneria spiralis*

In the present research, *V. spiralis* was collected from Fengcheng, a county located by the Ganjiang River from which water will flow into Poyang Lake through a 20 km distance, Jiangxi in March, 2016. After two weeks preculture and acclimation in the laboratory with dechlorinated municipal water with the addition of Hoagland Nutrient Solution at a proportion of 1 : 10 (Beijing Kolaibo Technology Ltd Co.), well-grown plants were rinsed with tap water and distilled water and dried by filter paper. Uniform height samples with $1.19 \pm 0.12$ g weight and $10 \pm 0.45$ cm height were chosen as experimental species and planted in sandy loam soil from April to September, 2016. To simulate natural conditions, this experiment was carried out outdoors in a field of the YIFU experimental building of Nanchang Institute of Technology (Nanchang, China). There was no other construction within 100 m of the site, and a nearby irrigation canal brought water from the Fuhe River. As shown in figure 2, the designed plastic

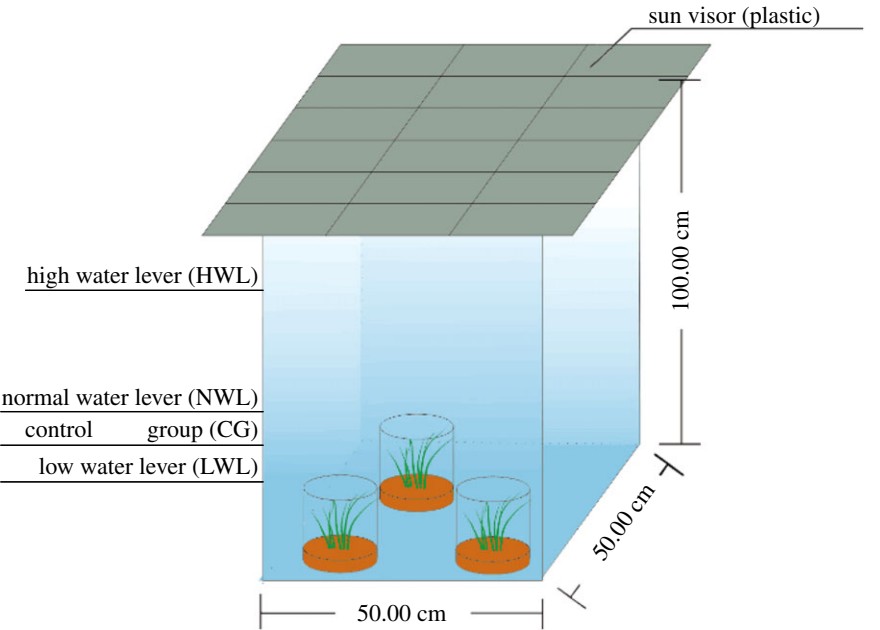

**Figure 2.** Experimental device simulating historical levels of low, normal and high wetland plant submersion in Lake Poyang, China.

bucket with $10 \pm 0.5$ cm of sandy loam soil at the bottom was placed in a glass tank with length 50 cm × width 50 cm × height 100 cm. The 12 strains of precultured *V. spiralis* were moved from the laboratory into this equipment. Throughout the experiment, the water in the tank was half replaced to maintain water quality and transparency with dechlorinated municipal water with the addition of Hoagland Nutrient Solution at a proportion of 1 : 10 (Beijing Kolaibo Technology Ltd Co.) every week.

## 2.2. Hydrological data

Based on the long-term hydrological data from Poyang Lake, Tan *et al.* [22] described the discipline of the annual water level and summarized the interannual variation characteristics of disc lakes located in the areas from the National Nature Reserve in Wu Cheng. Three typical hydrological years from the most recent 10 years, named NWL years (2009 and 2013), high water level (HWL) years (2010 and 2012) and an LWL year (2011), were chosen for simulation. The results of transparency research showed that the transparency in the LWL year ranged from 40 to 60 cm, and the maximum value was 80 cm. From March to June in HWL years, the lake water transparency fluctuated from 15 to 40 cm and could reach 40 to 80 cm in late July. The research results also showed that the water depth in NWL years ranged from 0.50 to 1.50 m from April to June and could reach 2.5 to 3.0 m after July. In HWL years, the water depth in disc lakes could rise rapidly to a depth of 2 m or more, and it continued to rise in late September, with the water depth ranging from 2.5 to 4 m.

## 2.3. Experimental design

To reflect the hydrological characteristics of disc lakes, which usually refers to the lower area separated from the main lake area and formed as individual internal lakes in the dry season in National Nature Reserve in Poyang Lake (Wu Cheng), the maximum water level in the HWL group, the minimum water level in the LWL group and the average monthly water level in the NWL group were used for laboratory simulation. The hydrological data from 2009 were used as the control group (CG) for *V. spiralis*, which grew normally and completed its life history in this year [22]. The detailed data for this experiment are listed in table 1. Owing to the difficulty in simulating the high water depth, the equipment was designed based on the theory described by the following equations [23]. By the mentioned method, the water depth can be controlled by the control net.

$$\mathrm{ST} = \frac{-\ln(0.2)}{K}, \tag{2.1}$$

and

$$I_{\mathrm{Z}} = I_{\mathrm{O}}\mathrm{e}^{-kh}, \tag{2.2}$$

**Table 1.** The water depth gradient design in different typical years.

|  | April | May | June | July | August | September |
|---|---|---|---|---|---|---|
| High flow year | 3.0 m | 3.5 m | 4.0 m | 4.5 m | 4.0 m | 3.0 m |
| Normal flow year | 0.5 m | 1.0 m | 1.5 m | 2.0 m | 2.5 m | 1.0 m |
| Low flow year | 0.3 m | 0.5 m | 1.5 m | 0.8 m | 0.7 m | 0.7 m |
| Control group | 0.5 m | 0.8 m | 1.0 m | 1.5 m | 2.0 m | 0.8 m |

where ST represents the transparency (cm); $K$ is the optical attenuation coefficient; $I_O$ represents light intensity at a 1 m point under the surface (lux); $h$ is the water depth (cm); the depth of the net will be below 1 m; and $I_Z$ represents the light intensity under the water at a depth of $h$.

## 2.4. Plant physical analysis

The light intensity at the plant's top on sunny, cloudy and rainy days was determined using the portable underwater radiometer (ZDS-10 W-2D, Shanghai Jiahe instrument Co., Ltd) at 12.00 every day. At the end of every month, three plant samples were taken in each parallel group and rinsed with tap water and distilled water, dried by filter paper and transported to the laboratory for physical measurements. The morphological parameters (plant height and biomass) of *V. spiralis* were then measured; plant height was measured by a measuring scale, and the biomass was determined by weighing. The average height increment of vegetation is defined as the difference in plant height between two adjacent months, and it is calculated from subtracting the average height of plants in the next month by the average height in the previous month. After that, leaves were cut separately into small pieces to a size of less than 1 mm by stainless steel scissors and then separately packed into a polyethylene Ziploc bag, marked and numbered and then stored in the refrigerator at $-40°C$ for biomarker assays.

## 2.5. Biomarker assays

Leaf samples (0.5 g) were accurately weighed, fully ground within nine volumes of cold buffer (0.15 M KCl, 0.1 M Tris−HCl, pH 7.4) and centrifuged for 25 min (12 000$g$) at 4°C. Supernatants were used as the extract for the protein content and enzymatic activity determination. Reagent Kits provided by Nanjing Jiancheng Biological Technology Co. Ltd were used for soluble protein contents and the enzyme activities analysis, including SOD, POD and CAT activities, and lipid degradation product presented by TBARS contents. The hydroxylamine method (xanthine oxidase method) was used to determine the SOD activities (U mg$^{-1}$ protein) at 550 nm with a microplate spectrophotometer (SpectraMax M5, Molecular Devices, USA) [24]. POD activities (U mg$^{-1}$ protein) could be determined by catalysing the corresponding substrate at 420 nm. The ammonium molybdate and hydrogen peroxide react and produce a result by which CAT activities (U mg$^{-1}$ protein) could be measured at 405 nm. As a biomarker, TBARS (nmol mg$^{-1}$ protein) is a lipid degradation product and could be determined using thiobarbituric acid (TBA), which is a red product and has a maximum absorption peak at 532 nm.

## 2.6. Statistical analysis

Beliaeff & Burgeot [14] mentioned a method in 2002 that can combine all the measured biomarker responses into one general stress index. In the present study, the growth of plant height, aboveground and underground biomass, and SOD, POD and CAT activities are combined into an IBR comprehensive index after standardization. Using this method, the comparison between different experimental groups can be accessed and the visualization results will be more acceptable.

The data were presented as the mean value $\pm$ standard deviation. The differences between the groups were tested using the LSD method in a one-way ANOVA by IBM SPSS 22.0, in which significant difference was indicated with $p < 0.05$ and highly significant difference with $p < 0.01$. This study uses Microsoft Excel 2010 software to do the statistics and data calculations. Adobe Photoshop CS6 and Corel DRAW are used for image optimization.

rsos.royalsocietypublishing.org    R. Soc. open sci. **5**: 180729

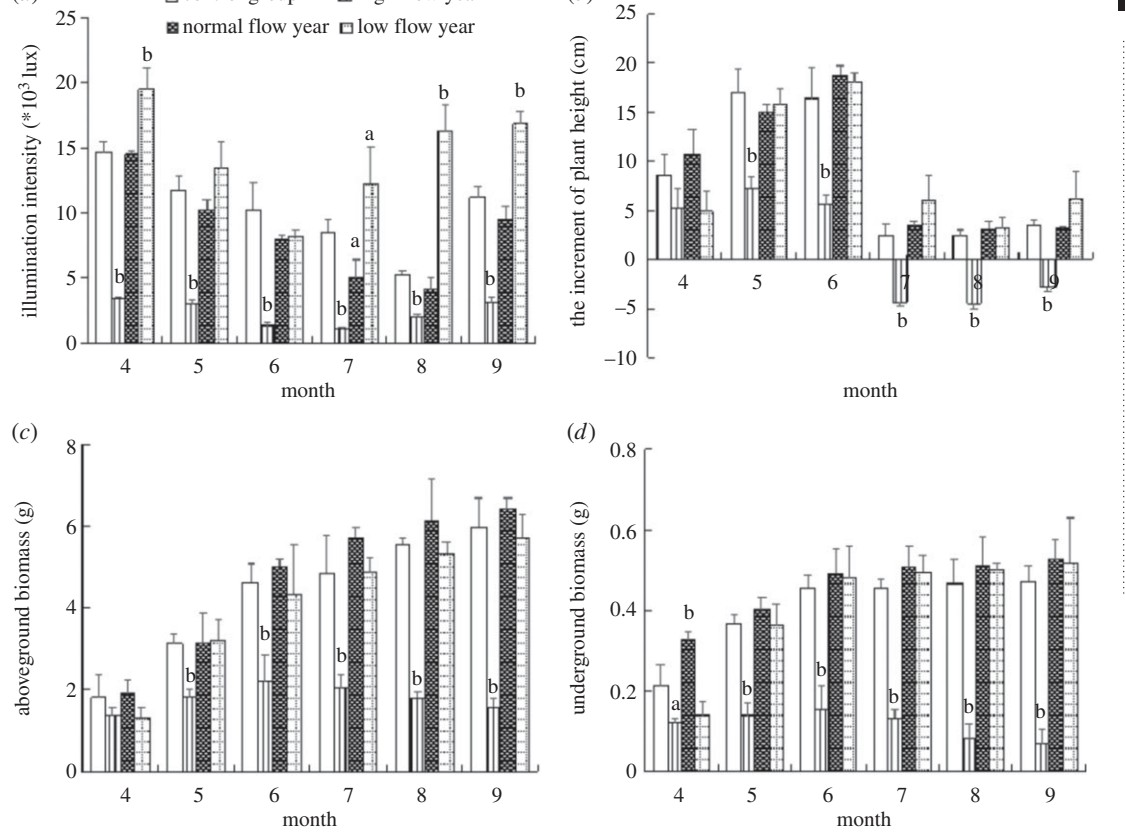

**Figure 3.** Response of light intensity (*a*), plant height increment (*b*), aboveground biomass (*c*) and underground biomass (*d*) at typical hydrological processes. The lower case a and b indicate values that are significantly different with control values (a indicating $p < 0.05$, b indicating $p < 0.01$).

## 3. Results

### 3.1. Light intensity and morphology characteristics

Under the simulation for typical hydrological processes of Poyang Lake, the underwater light intensity, eelgrass plant height growth, ground biomass and belowground biomass of *V. spiralis* showed a significant change pattern. As shown in figure 3*a*, the underwater light intensity at the top of the plant for all groups gradually decreased from April to July and increased gradually after August, and a 'V' shape is observed for the change pattern. In the HWL group, the underwater light intensity is always lower compared with the other groups, and the minimum value is observed in July of $1.14 \times 10^3$ lux. Compared with the CG, the underwater light intensity in the HWL group always has a highly significant difference ($p < 0.01$). The underwater light intensity in the NWL group showed a minimum value in August of $4.08 \times 10^3$ lux. In general, no significant difference in underwater light intensity was observed after comparison with the CG. For the LWL group, the underwater light intensity will reach the lowest value of $8.1 \times 10^3$ lux in June. Not including the measured values in May and June, the underwater light intensity in other months also showed significant differences compared with that of CG at the same time ($p < 0.05$ or $p < 0.01$).

As shown in figure 3*b*, not including the HWL group, the plant height increment in different groups increases gradually from April to June and maintains a steady growth in the later stage. The maximum value of the plant height increment for the NWL and LWL are determined in June with average values 18.22 and 17.32 cm. In general, no significant differences are observed for plant height increment among CG and NWL and LWL. Instead, the plant height increment for HWL shows significant differences compared with other groups ($p < 0.01$). A negative plant height increment is even observed from July to September, which is usually referred to the flooding period of Lake Poyang. Compared with CG, the plant height increment in May and June for HWL are 66.1% and 81.7% lower during the same period.

rsos.royalsocietypublishing.org    R. Soc. open sci. **5**: 180729

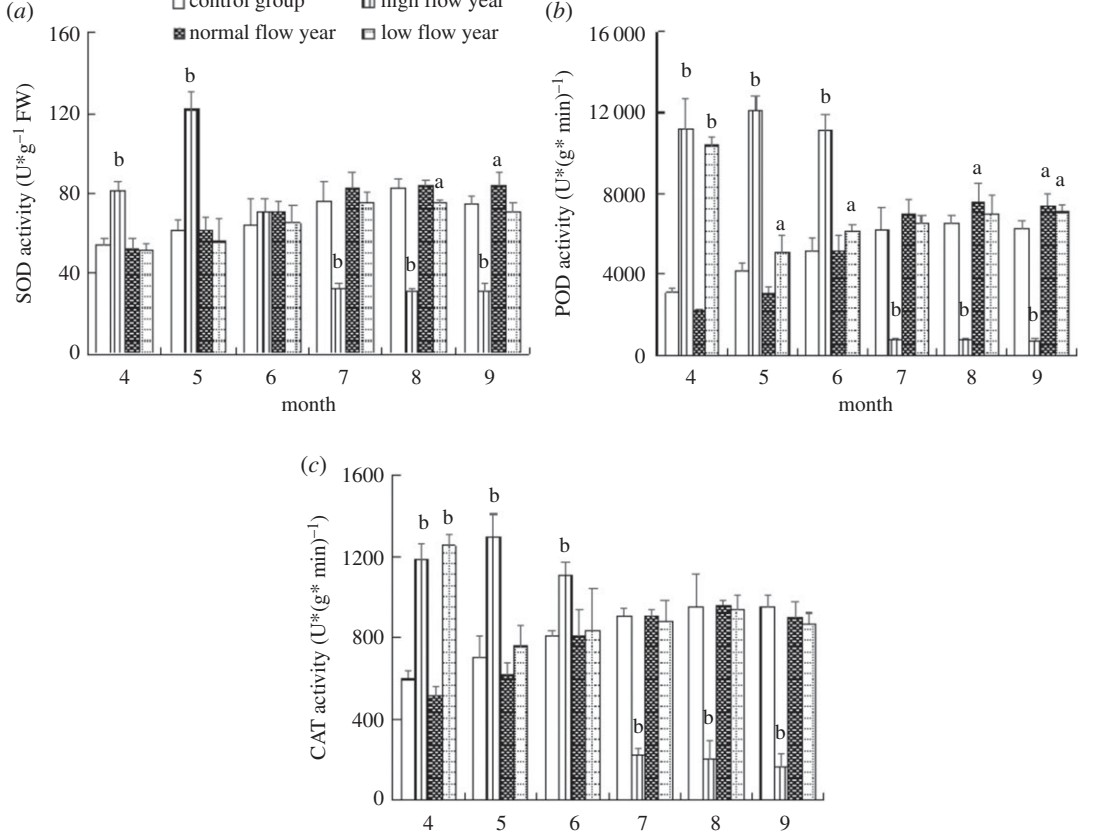

**Figure 4.** Responses of SOD activity (*a*), POD activity (*b*) and CAT activity (*c*) measured in the leaves at the typical hydrological processes. The lower case a and b indicate values that are significantly different than background values (a indicating $p < 0.05$, b indicating $p < 0.01$).

Biomasses from aboveground and belowground are shown in figure 3*c,d*. In conclusion, both aboveground biomass and belowground biomass from NWL and LWL increase gradually in the whole growth period and reach the highest value in September, with 6.42 and 5.73 g for aboveground biomass, respectively. Compared with the CG, there is no significant difference for both aboveground and belowground biomass between NWL, LWL and CG. Similar to the plant height increment, both aboveground and belowground biomass for HWL begin to decrease gradually from June after a slow increase in the first three months and show significant differences compared with the CG ($p < 0.01$).

## 3.2. Response of antioxidant enzymes

SOD, POD and CAT activities in leaves of *V. spiralis* are presented in figure 4. Taking the entire hydrological process into consideration, the responses of the three biomarkers measured for the CG show a slow, steady growth trend from April to September. As shown in figure 4*a*, the activity of SOD could reach the highest values in July and August, with 84.3 U g$^{-1}$ and 75.4 U g$^{-1}$, respectively. POD and CAT activities of *V. spiralis* are shown in figure 4*b,c*. For the CG, they could reach the maximum values of 7593.5 U (g min)$^{-1}$ and 956.5 U (g min)$^{-1}$ in August, respectively.

In general, apart from April, the enzymatic activities of NWL and LWL do not show significant differences compared with that of the CG. However, the biomarker activities of HWL change significantly with the change of hydrological processes. As for HWL, SOD, POD and CAT activities are usually induced at the initial stages and significantly inhibited in different degrees in the later stages. Among the three antioxidant enzymes, SOD activities are significantly induced ($p < 0.01$) in April and May, and the biggest inducement is observed in May. However, a significant reduction ($p < 0.01$) is also found from July to September when Lake Poyang is experiencing the flooding season. Similar to SOD activities responses, POD and CAT enzymes are also observed with the same

rsos.royalsocietypublishing.org    R. Soc. open sci. **5**: 180729

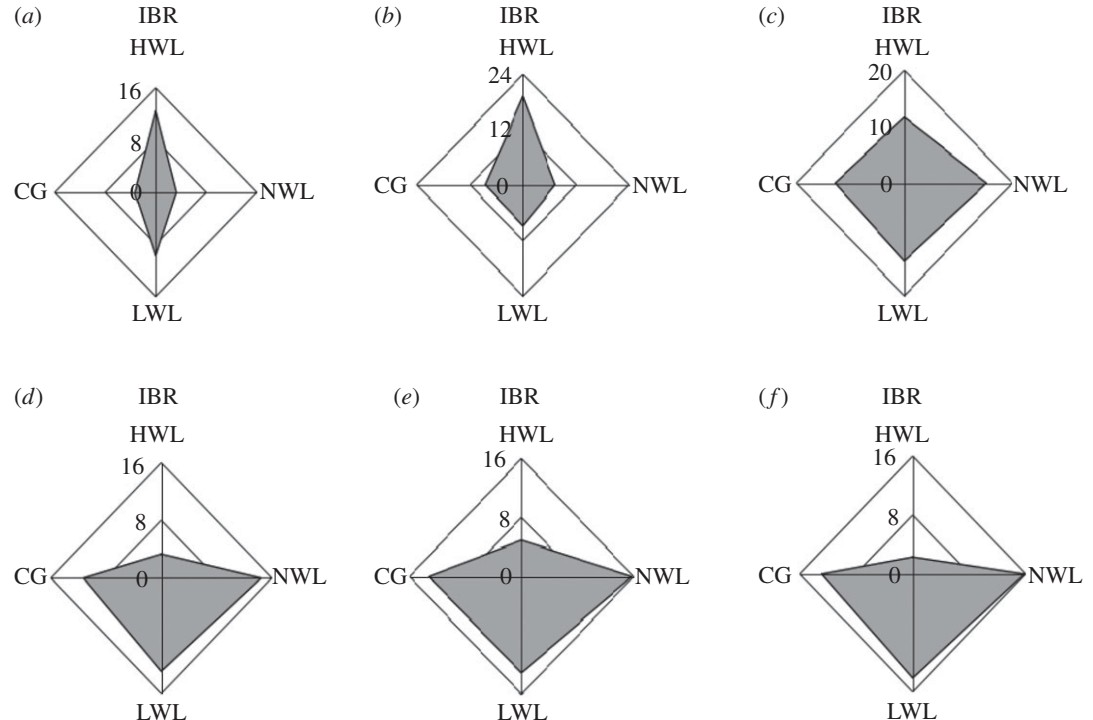

**Figure 5.** Integrated biomarkers responses for April–June (*a*–*f*) under three different typical hydrological processes (high water level (HWL), normal water level (NWL), low water level (LWL), and control group (CG)).

trend ($p < 0.01$). When the water level in Lake Poyang is rising slowly, these enzymes are significantly induced in April, May and June. Furthermore, POD and CAT activities for HWL are significantly different ($p < 0.01$) from CG from July to September. The minimum values of SOD, POD and CAT activities for HWL are observed in September.

## 3.3. Integrated biomarker responses

For the entire typical hydrological process, figure 5*a*–*f* is used to indicate IBR values calculated from April to September with a radar map. IBR values are shown by the radius coordinates, and different hydrological groups are listed in four directions, defined as HWL, NWL, LWL and CG, individually. In general, a large range variation for IBR values is observed for different hydrological stages and different experimental groups. Star plots in figure 5*a,b* for IBR values refer to the integrated responses during the slowly rising period when the Lake Poyang is collecting rainwater from the whole basin. The IBR values exhibit obvious spatial changes that are consistent with the responses of growth characteristics and enzymes responses of *V. spiralis*. Figure 5*d*–*f* refers to the distribution of IBR values for the flooding season, when Lake Poyang is experiencing a period of high water levels. As mentioned previously, growth characteristics and enzymes responses of *V. spiralis* from HWL years in this period show significant inhibition.

## 4. Discussion

As one of the most direct and basic indicators, physical growth could reflect the growth characteristics of the plant during different stages. Public reports [25] indicated that each plant had its own growth 'window of opportunity', which meant different plants had special suitable water levels. As the energy source, the light intensity has a key influence on plant photosynthesis and directly affects the plant growth. In this study, the growth of *V. spiralis* from an HWL year was significantly inhibited, and the growth of plant height was even negative in the hydrological flooding period. As presented in figure 3*a*, the intensity of light decreases as water depth increases from April to July, and it increases as the water level drops in August and September. Accordingly, plant height increased gradually in the first stage from April to July and remained steady at the late stage in August and

rsos.royalsocietypublishing.org   R. Soc. open sci. 5: 180729

September with the increase and decrease in water level (figure 3b). For HWL, the plant height reached the maximum value when the light intensity was $2.01 \times 10^3$ lux, and there is a negative correlation between the plant height and light intensity from April to July. There was a significant negative correlation with a correlation coefficient of $R^2 = 0.9666$ between the average light intensity underwater and average height of vegetation growth performed with a polynomial regression analysis using data from April to July ($p < 0.001$).

Biomass is one of the main morphological indices to measure the growth state of plants, and the change of biomass directly reflects the growth status of plants. As presented in figure 3c,d, plant growth and biomass are significantly inhibited when the light intensity is lower than the growth requirement or plants suffer long-term higher water levels. For HWL, the aboveground and belowground biomass reached the maximum values when the light intensity was $1.98 \times 10^3$ and $2.04 \times 10^3$ lux, respectively, and then began to wither later. This result is consistent with the conclusion that the biomasses of the community of Dioscorea microphylla and spike foxtail were significantly lower in the deep water area than those measured in the shallow water area [26]. According to the statistical results, Johnson et al. [27] also made a similar conclusion for the significant reduction of vegetation biomass in the continuous high water stress in the Orchichun wetland of the United States. A polynomial regression analysis from data from April to July indicated a negative correlation between the plant aboveground average biomass, belowground average biomass and average light intensity, with correlation coefficients $R^2 = 0.9915$ and $0.8773$, respectively ($p < 0.001$).

Strand & Weisner [28] designed a field experiment with submerged plant Cercospora spp. to verify the effect of water depth on plant growth. This study revealed that Cercospora spp. could fully cope with the stress of light and adapted well within the water range. The normal growth of Cercospora spp. would not be affected under a short duration of a higher water level due to the strong tolerance. For the NWL and LWL, as shown in figure 3b–d, the biomass of V. spiralis gradually increased, although the intensity of underwater light was decreasing. Similar to the CG, the biomass increase in May and June was obviously higher than that in other months. For the NWL, the plant height of submerged plants was always increasing and the closest to that of the CG, especially in May and June, with the maximum increase in the range of $8.05–13.47 \times 10^3$ lux. For water chestnut, the plant height increased with the increase in water level when the water depth ranged from 7 to 54 cm [29]. The results of water depth gradient experiments on Scirpus triqueter also revealed that the plant height increased with the increase in water level [30]. Similar to NWL, the rapid growth of plant height in LWL occurred in May and June, too.

As shown in figure 6b–d, the biomass distribution characteristics between the aboveground and belowground biomass of the NWL and LWL indicated that more biomass of V. spiralis would be distributed to the aboveground part as the water level increased, which may be due to the weakening of underwater light intensity and the fact that more aboveground biomass could receive more light. This result was consistent with the results conducted in 2014 with the emergent aquatic plant Cyperus rotundus and the submerged plant eel grass [31]. This study [31] stated that plants would allocate more biomass to the aboveground part to receive more light and more favourable photosynthetic and respiratory conditions. However, other studies on the relationship between the aboveground and belowground distribution of biomass showed that the ratios of Carex and New Zealand hemp were not correlated with the water level [32,33].

The results showed that the oxygen received by plants would decrease and the reactive oxygen species would continuously accumulate when submerged plants were under the deep water stress [34]. In this case, the antioxidant systems in plant leaves, including SOD, POD and CAT, could remove excess reactive oxygen to maintain the balance of reactive oxygen species. As illustrated in figure 4a–c, the activities of the three antioxidant enzymes in plant leaves for different treated groups increased continuously with the rise of water level and reached the maximum in August. This result is consistent with the antioxidant enzyme responses of the emergent aquatic plant Acorus calamus in a water depth experiment [35]. For LWL, POD and CAT activities were significantly induced in April ($p < 0.01$) when the water level was continuously low. Under the high intensity of light, POD and CAT were more easily induced to eliminate the excess reactive oxygen caused by strong light stress. By contrast, the antioxidant enzymes in HWL were significantly inhibited, which was probably due to the accumulation of active oxygen species under the condition of the HWL for a long time and the insufficient protection of the enzyme system to remove extra active oxygen.

The integrated biomarkers comprehensive index (IBR) that combined all biomarkers together could fully take advantage of different biomarkers to reliably interpret the effect of toxicological examinations, while this method could also avoid different evaluations with only one biomarker under the effect of outer stress. The IBR index mentioned by Beliaeff & Burgeot [14] has been widely

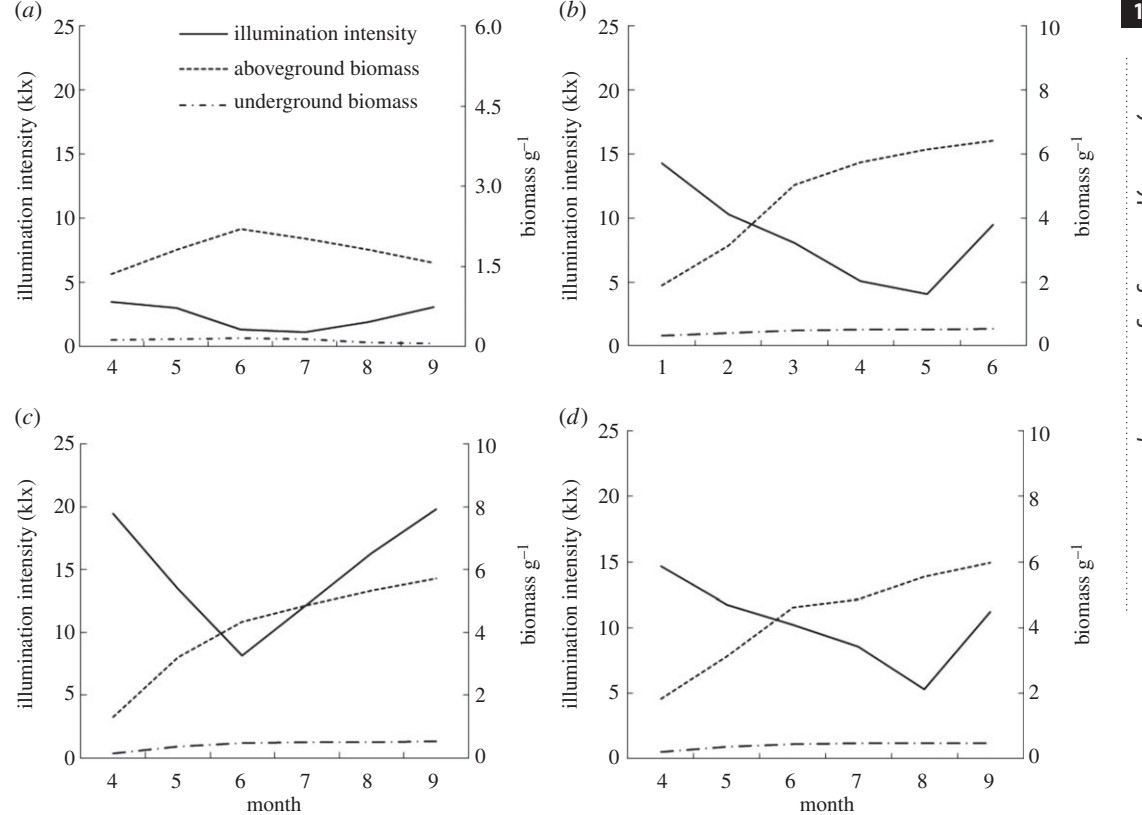

**Figure 6.** The relationship between light and biomass (HWL (*a*), NWL (*b*), LWL (*c*), Control Group (*d*)).

accepted to assess integrated effects in oceans [36] and freshwater lakes [37]. As presented in figure 5*a*–*f*, plant morphological characteristics and antioxidase stress parameters of *V. spiralis* were chosen to provide an integrated evaluation of different hydrological processes. The integrated biomarker responses among different experimental groups were presented in the same star radar plots, which made the comparison among different groups more visible. Based on the IBR comparison results, the HWL group was observed with significant biomarker responses, which meant the higher water depth caused serious stress for plant growth. In fact, most of the *V. spiralis* died in the flooding season for the HWL group. The IBR index has also been proved as a useful evaluation method for the evaluation of ZnO NPs (nanoparticles) toxicity [38] and the assessment of field-contaminated soils [36]. In the Le'An Basin, polluted by heavy metals, Ji [13,39] revealed that pollutants measured from locally consumed vegetables and paddies showed a strong agreement with the integrated biomarker response (IBR).

# 5. Conclusion

This study, by designing a simulation experiment, aimed to evaluate the effects of different hydrological processes on the growth of the submerged plant *V. spiralis* in Lake Poyang by combining physical indicators and biological indicators from three typical hydrological processes. The results clearly revealed that hydrological processes significantly contributed to plant growth and that submerged plants balanced outer stress via antioxidase protection mechanisms. The SOD, POD and CAT activities could be used to evaluate the stress from higher water depths. In combination with morphological characteristics and biomarkers, integrated biomarkers responses (IBR) showed strong coordination between the water level and biomarker responses in different months.

Data accessibility. The dataset supporting this article has been deposited at the Dryad Digital Repository: http://dx.doi.org/10.5061/dryad.09hh050 [40].

Authors' contributions. Y.J. and S.Z. designed the study. Z.Y. and X.W. prepared all samples for chemical analysis and participated in data analysis. J.Z. collected and analysed the data. J.L. and L.X. coordinated the study and helped draft the manuscript. All authors gave final approval for publication.

Competing interests. The authors declare no competing interests.

Funding. This work was supported by Jiangxi Provincial Education Department (GJJ161094), National Natural Science Foundation of China (51469017; 51579127; 51769015; 51779005) and Jiangxi Provincial Technology Department (20171ACB21050).

Acknowledgements. We express thanks to Mrs Gao of Water Treatment Laboratory, NIT, for her assistance with data analyses and promotion of simulation experiment.

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
