## [Reviewer comments · Royal Society Open Science]

Review History

RSOS-180729.R0 (Original submission)

Review form: Reviewer 1

Is the manuscript scientifically sound in its present form?

No

Are the interpretations and conclusions justified by the results?

Yes

Is the language acceptable?

No

Is it clear how to access all supporting data?

Not Applicable

Do you have any ethical concerns with this paper?

No

Have you any concerns about statistical analyses in this paper?

Yes

Recommendation?

Major revision is needed (please make suggestions in comments)

Comments to the Author(s)

Review of Ji et al. "Integrated biomarker responses of submerged macrophyte *Vallisneria spiralis* with the hydrological process from Lake Poyang, China"

General comments

This manuscript reports the effects of different levels of flooding on the growth of and antioxidant enzyme activity within the wetland plant *Vallisneria spiralis*. The objective of this study was to use a laboratory experiment simulating historical levels of low, normal, and high wetland plant submersion in Poyang Lake, China to understand how light intensity, plant height, plant biomass, and antioxidant enzyme activity respond to these different hydrologic conditions over six months. The authors found that light intensity was different from the control in both the low and high water level treatments, but that plant height, aboveground, and belowground biomass were only different from the control in the high water level treatment. Biomarker responses seem to only have responded to the high water level treatment as well, but the meaning of these different biomarker responses is difficult to interpret as the reader.

The results of this study are interesting and contribute to our understanding of how wetland plants such as *Vallisneria spiralis* respond to variation in hydrologic conditions. This work is important to understanding of how the Three Gorges Reservoir is changing downstream ecosystems.

Although this work contributes important information as to how *Vallisneria spiralis* growth and antioxidant enzyme activity within the plant respond to water level fluctuations, there are many key details that are missing, particularly in regard to the hypothesis, methods, and data analysis. Here are some major suggestions:

- 1) While the introduction does a good job of setting up the importance of the study, the authors could do a better job of explaining what their predictions or hypotheses were for the experiment.
- 2) The methods need much more detailed information. Were the glass jars (tanks?) outside, in a greenhouse, or indoors? Was water temperature measured? Where was the water that was in the tanks from? At what frequency did the authors add nutrients to the water? On what plant material did the authors conduct the biomarker assays? Were plants destructively sampled to determine biomass, or were they weighed while still wet? Etc... The authors should write the methods so that a reader would have enough information to exactly replicate the study.
- 3) There is no explanation of which statistical tests the authors used beyond generally stating that Excel was used. It is very important that the authors choose the appropriate analysis for tests among groups, especially if the data are not normally distributed.
- 4) I would also recommend that the authors pay particular attention to improving the grammar in their manuscript. Notably, much of the manuscript was written in the future tense. Given that

this experiment was already conducted, the authors should never use the future tense and instead use the past tense (or present tense where appropriate).

With major revisions, the authors should be able to clarify the missing information in the methods and reduce grammatical errors in the writing. This study should be of interest to a broad audience given the increasing construction of dams around the globe that are having downstream impacts on the hydrologic conditions experienced by wetland plants.

Specific comments:

Note: I will not be correcting all of the grammatical mistakes in the manuscript. Instead, I aim to provide comments on the substance of the writing.

ABSTRACT

Line 2: Add “,China” after Lake Poyang

Line 16-17: Do not use acronyms in abstracts without defining them

End: Try to add a general sentence that brings all of the results (including biomass, antioxidant enzyme activities, etc.) together into a main final conclusion of the study -- Something like “We found that high flooding levels had the strongest negative effect on growth and enzyme activity of *Vallisneria spiralis*”

INTRODUCTION

Page 3, Line 29: What do the authors mean by “As the energy source of photosynthesis, rhythmic changes in wetland water level can effect...”? Do they mean to say that water is affecting the light levels? The way the sentence is written as of now it sounds like the authors are saying that water is the energy source

Page 4, Line 22: It would be helpful for there to be a map of where the lake is in China, maybe in the methods?

Page 5, Line 18: Start a new paragraph at “In order to fully...”

METHODS

Page 5, Line 54: Does this weight include roots or the leaves or the whole plant? Is it a wet weight? If so, how did you control for differences in the amount of water on and in the plant?

Page 6, Line 3: Do not use the future tense, here or nearly anywhere else

Section 2.3: This section need the most work. The explanation of the experimental setup is confusing and needs many more details related to how the experiment was run for 6 months.

What is a disc lake?

Page 7, Line 35: Remove “accurately”. Every measurement has error associated with it

Page 7, Line 43: Redefine enzyme abbreviations

Page 7, Line 45: Provide citations for the hydroxylamine method

What instrument were the activities measured on?

Section 2.6 – This is another section that needs a lot of work. Which statistical analyses were used to test what? Describe all statistical analyses in this section.

RESULTS

Section 3.1

What does klx stand for?

How did you test for a significant difference? Please report the statistical method and the P-value

I am confused by the metric plant height increment. This was not explained in the methods section

Biomass should be normalized to area. Is this biomass within the jars? If so, were the plants destructively sampled? This also needs more explanation in the methods

Section 3.2

The sentence “IBR values reflect the comprehension responses to the stress of anoxia condition” should be moved to the data analysis section (and the grammar should be checked). Instead, since this is the results, the authors should describe the differences among treatments in these star plots

How did the authors assess significant inhibition?

Section 3.3

Much of this section should be moved to the methods because it is simply describing how IBR was used, and not what the results were.

DISCUSSION

How do you disentangle the impact of light intensity versus anoxic conditions?

Page 11, Line 37: This correlation is not shown in Figures 2A and 2B, but it is good to see some reported regression values. What is the p-value of this relationship? It clearly is significant if it explained so much of the variance

Were the raw data or was the median value used for the regression analysis?

Do not use the future tense. You are discussing work that has already been done.

Page 11, Line 54: “above ground” and “below ground” should both be one word (e.g. “aboveground”)

Page 13, Line 18: This would be a good sentence to incorporate into the end of the abstract because it does a good job of summarizing all of the results together

Page 14, Line 22: Define NP to nanoparticles

Page 14, Line 27: How did you assess “Well agreement”?

FIGURES

Figure 1. Needs to have a scale and much more information about the device, especially given that the water fluctuated.

Figure 2.

What is the random "5.29" doing in plot 2A?

The standard deviation bars are remarkably consistent from month to month. Are the authors certain that they calculated a separate standard deviation for each month?

How did the authors test for significant differences?

Figure 3

How did the authors test for significant differences?

Figure 4

Make sure that text from separate plots does not overlap in the final figure

"CK" in the plots should be changed to "CG"

Figure 5.

- The legend only needs to be shown once for all four panels, preferably to the right of all four plots.

- The scale needs to be the same magnitude among all four panels (e.g. Illumination intensity needs to go from 0 to 25 on all 4 plots)

- This is a really interesting figure!

Table 1.

- "may" need to be capitalized to "May"

Review form: Reviewer 2

Is the manuscript scientifically sound in its present form?

Yes

Are the interpretations and conclusions justified by the results?

Yes

Is the language acceptable?

Yes

Is it clear how to access all supporting data?

Yes

Do you have any ethical concerns with this paper?

No

Have you any concerns about statistical analyses in this paper?

No

Recommendation?

Accept with minor revision (please list in comments)

Comments to the Author(s)

Special comments:

1. The paper's title could be revised as "Integrated Biomarker Responses of submerged macrophyte *Vallisneria spiralis* for the hydrological process of Lake Poyang, China".
2. Page 4, line 22, change "square kilometer" to km².
3. Page 4, line 48, change "Poyang lake" to "Poyang Lake".
4. Page 5, line 2, change "biochemical of *Vallisneria spiralis*" to "biochemical indices of submerged plants".
5. "*Vallisneria spiralis*" can be abbreviated as "*V. spiralis*" after the first use.
6. Page 6, line 11, Tan[22] should be Tan et al. [22]. Same errors in Page 12 lines 9 and 11
7. Page 6, line 41, change 0.5cm to 0.5 cm. Pay more attention the essential space in the full text.
8. Page 7, line 7, change "The water depth" to "the water depth".
9. Page 7, line 43-45, revise this sentence "enzyme activities analysis, including SOD, POD, CAT and TBARS". TBARS is not enzyme.
10. Page 6, line 47, Materials and Methods, authors should give an explanation what is "disc lake" and what is the relationship between the "disc lake" and experimental design?
11. Page 8, line 11-13, IBR calculation need to provide some details. Not all physical parameters and biomarkers were used?
12. Page 11, line 37, an explanation of data analysis should be provided for "correlation coefficient square R² 0.9666".
13. Authors used full text "Figure" in text, while used abbreviation "Fig." in attached documents. Please check and keep the same description in the whole paper whatever in the diagram or in the text.
14. Fig. 4 was not clearly presented, Please check and revise.
15. Taking higher frequent used "High Water Level or High Water Year" for consideration, authors should mention the abbreviations appeared at the first time in text and then used abbreviation in the later description. Please Check and revise.
16. Tables 1, please supply an explanation in contents why this experimentation only simulates 6 months, which is little different in description as mentioned in introduction.
17. The language used in this paper could be further improved by native English speakers or professional English editing institutes.

Decision letter (RSOS-180729.R0)

29-Aug-2018

Dear Dr Ji,

The editors assigned to your paper ("Integrated Biomarker Responses of submerged macrophyte *Vallisneria spiralis* with the hydrological process from Lake Poyang, China") have now received comments from reviewers. We would like you to revise your paper in accordance with the referee and Associate Editor suggestions which can be found below (not including confidential reports to the Editor). Please note this decision does not guarantee eventual acceptance.

We recommend that you ask a native speaker of English or solicit the support of a language polishing service (<https://royalsociety.org/journals/authors/language-polishing/>) prior to resubmitting the manuscript.

Please submit a copy of your revised paper before 21-Sep-2018. Please note that the revision

deadline will expire at 00.00am on this date. If we do not hear from you within this time then it will be assumed that the paper has been withdrawn. In exceptional circumstances, extensions may be possible if agreed with the Editorial Office in advance. We do not allow multiple rounds of revision so we urge you to make every effort to fully address all of the comments at this stage. If deemed necessary by the Editors, your manuscript will be sent back to one or more of the original reviewers for assessment. If the original reviewers are not available, we may invite new reviewers.

- Data accessibility

<http://datadryad.org/submit?journalID=RSOS&manu=RSOS-180729>

- Competing interests

- Authors' contributions

All submissions, other than those with a single author, must include an Authors' Contributions section which individually lists the specific contribution of each author. The list of Authors should meet all of the following criteria; 1) substantial contributions to conception and design, or

acquisition of data, or analysis and interpretation of data; 2) drafting the article or revising it critically for important intellectual content; and 3) final approval of the version to be published.

- Acknowledgements

- Funding statement

Please note that Royal Society Open Science charge article processing charges for all new submissions that are accepted for publication. Charges will also apply to papers transferred to Royal Society Open Science from other Royal Society Publishing journals, as well as papers submitted as part of our collaboration with the Royal Society of Chemistry (<http://rsos.royalsocietypublishing.org/chemistry>). If your manuscript is newly submitted and subsequently accepted for publication, you will be asked to pay the article processing charge, unless you request a waiver and this is approved by Royal Society Publishing. You can find out more about the charges at <http://rsos.royalsocietypublishing.org/page/charges>. Should you have any queries, please contact openscience@royalsociety.org.

on behalf of Dr Punidan Jeyasingh (Associate Editor) and Prof. Jon Blundy (Subject Editor)
openscience@royalsociety.org

Associate Editor's comments (Dr Punidan Jeyasingh):

This manuscript reports results exploring the effects of hydrology on the performance of *Vallisneria*. The manuscript was reviewed by two experts. Both of whom were enthusiastic about the work (so was I during pre-assessment). Both reviewers raise several important issues that need to be addressed. I felt the comments were fair and constructive. In addition, and more importantly, the manuscript suffers from numerous grammatical issues. I urge the authors to consult an English language editing service (or an Anglophone colleague) before resubmission. This is a very nice piece of work that needs to be communicated more effectively. With much gratitude to the expert reviewers, I invite the authors to make these revisions.

Comments to Author:

Reviewers' Comments to Author:

Reviewer: 1

Comments to the Author(s)

Review of Ji et al. "Integrated biomarker responses of submerged macrophyte *Vallisneria spiralis* with the hydrological process from Lake Poyang, China"

General comments

This manuscript reports the effects of different levels of flooding on the growth of and antioxidant enzyme activity within the wetland plant *Vallisneria spiralis*. The objective of this study was to use a laboratory experiment simulating historical levels of low, normal, and high wetland plant submersion in Poyang Lake, China to understand how light intensity, plant height, plant biomass, and antioxidant enzyme activity respond to these different hydrologic conditions over six months. The authors found that light intensity was different from the control in both the low and high water level treatments, but that plant height, aboveground, and belowground biomass were only different from the control in the high water level treatment. Biomarker responses seem to only have responded to the high water level treatment as well, but the meaning of these different biomarker responses is difficult to interpret as the reader.

The results of this study are interesting and contribute to our understanding of how wetland plants such as *Vallisneria spiralis* respond to variation in hydrologic conditions. This work is important to understanding of how the Three Gorges Reservoir is changing downstream ecosystems.

Although this work contributes important information as to how *Vallisneria spiralis* growth and antioxidant enzyme activity within the plant respond to water level fluctuations, there are many key details that are missing, particularly in regard to the hypothesis, methods, and data analysis. Here are some major suggestions:

- 1) While the introduction does a good job of setting up the importance of the study, the authors could do a better job of explaining what their predictions or hypotheses were for the experiment.
- 2) The methods need much more detailed information. Were the glass jars (tanks?) outside, in a greenhouse, or indoors? Was water temperature measured? Where was the water that was in the tanks from? At what frequency did the authors add nutrients to the water? On what plant material did the authors conduct the biomarker assays? Were plants destructively sampled to determine biomass, or were they weighed while still wet? Etc... The authors should write the methods so that a reader would have enough information to exactly replicate the study.
- 3) There is no explanation of which statistical tests the authors used beyond generally stating that Excel was used. It is very important that the authors choose the appropriate analysis for tests among groups, especially if the data are not normally distributed.
- 4) I would also recommend that the authors pay particular attention to improving the grammar in their manuscript. Notably, much of the manuscript was written in the future tense. Given that this experiment was already conducted, the authors should never use the future tense and instead use the past tense (or present tense where appropriate).

With major revisions, the authors should be able to clarify the missing information in the methods and reduce grammatical errors in the writing. This study should be of interest to a broad

audience given the increasing construction of dams around the globe that are having downstream impacts on the hydrologic conditions experienced by wetland plants.

Specific comments:

Note: I will not be correcting all of the grammatical mistakes in the manuscript. Instead, I aim to provide comments on the substance of the writing.

ABSTRACT

Line 2: Add “,China” after Lake Poyang

Line 16-17: Do not use acronyms in abstracts without defining them

End: Try to add a general sentence that brings all of the results (including biomass, antioxidant enzyme activities, etc.) together into a main final conclusion of the study -- Something like “We found that high flooding levels had the strongest negative effect on growth and enzyme activity of *Vallisneria spiralis*”

INTRODUCTION

Page 3, Line 29: What do the authors mean by “As the energy source of photosynthesis, rhythmic changes in wetland water level can effect....”? Do they mean to say that water is affecting the light levels? The way the sentence is written as of now it sounds like the authors are saying that water is the energy source

Page 4, Line 22: It would be helpful for there to be a map of where the lake is in China, maybe in the methods?

Page 5, Line 18: Start a new paragraph at “In order to fully...”

METHODS

Page 5, Line 54: Does this weight include roots or the leaves or the whole plant? Is it a wet weight? If so, how did you control for differences in the amount of water on and in the plant?

Page 6, Line 3: Do not use the future tense, here or nearly anywhere else

Section 2.3: This section need the most work. The explanation of the experimental setup is confusing and needs many more details related to how the experiment was run for 6 months.

What is a disc lake?

Page 7, Line 35: Remove “accurately”. Every measurement has error associated with it

Page 7, Line 43: Redefine enzyme abbreviations

Page 7, Line 45: Provide citations for the hydroxylamine method

What instrument were the activities measured on?

Section 2.6 – This is another section that needs a lot of work. Which statistical analyses were used to test what? Describe all statistical analyses in this section.

RESULTS

Section 3.1

What does klx stand for?

How did you test for a significant difference? Please report the statistical method and the P-value

I am confused by the metric plant height increment. This was not explained in the methods section

Biomass should be normalized to area. Is this biomass within the jars? If so, were the plants destructively sampled? This also needs more explanation in the methods

Section 3.2

The sentence "IBR values reflect the comprehension responses to the stress of anoxia condition" should be moved to the data analysis section (and the grammar should be checked). Instead, since this is the results, the authors should describe the differences among treatments in these star plots

How did the authors assess significant inhibition?

Section 3.3

Much of this section should be moved to the methods because it is simply describing how IBR was used, and not what the results were.

DISCUSSION

How do you disentangle the impact of light intensity versus anoxic conditions?

Page 11, Line 37: This correlation is not shown in Figures 2A and 2B, but it is good to see some reported regression values. What is the p-value of this relationship? It clearly is significant if it explained so much of the variance

Were the raw data or was the median value used for the regression analysis?

Do not use the future tense. You are discussing work that has already been done.

Page 11, Line 54: "above ground" and "below ground" should both be one word (e.g. "aboveground")

Page 13, Line 18: This would be a good sentence to incorporate into the end of the abstract because it does a good job of summarizing all of the results together

Page 14, Line 22: Define NP to nanoparticles

Page 14, Line 27: How did you assess "Well agreement"?

FIGURES

Figure 1. Needs to have a scale and much more information about the device, especially given that the water fluctuated.

Figure 2.

What is the random "5.29" doing in plot 2A?

The standard deviation bars are remarkably consistent from month to month. Are the authors certain that they calculated a separate standard deviation for each month?

How did the authors test for significant differences?

Figure 3

How did the authors test for significant differences?

Figure 4

Make sure that text from separate plots does not overlap in the final figure
 “CK” in the plots should be changed to “CG”

Figure 5.

- The legend only needs to be shown once for all four panels, preferably to the right of all four plots.
- The scale needs to be the same magnitude among all four panels (e.g. Illumination intensity needs to go from 0 to 25 on all 4 plots)
- This is a really interesting figure!

Table 1.

- “may” need to be capitalized to “May”

Reviewer: 2

Comments to the Author(s)

Special comments:

1. The paper’s title could be revised as “Integrated Biomarker Responses of submerged macrophyte *Vallisneria spiralis* for the hydrological process of Lake Poyang, China”.
2. Page 4, line 22, change “square kilometer” to km².
3. Page 4, line 48, change “Poyang lake” to “Poyang Lake”.
4. Page 5, line 2, change “biochemical of *Vallisneria spiralis*” to “biochemical indices of submerged plants”.
5. “*Vallisneria spiralis*” can be abbreviated as “*V. spiralis*” after the first use.
6. Page 6, line 11, Tan[22] should be Tan et al. [22]. Same errors in Page 12 lines 9 and 11
7. Page 6, line 41, change 0.5cm to 0.5 cm. Pay more attention the essential space in the full text.
8. Page 7, line 7, change “The water depth” to “the water depth”.
9. Page 7, line 43-45, revise this sentence “enzyme activities analysis, including SOD, POD, CAT and TBARS”. TBARS is not enzyme.
10. Page 6, line 47, Materials and Methods, authors should give an explanation what is “disc lake” and what is the relationship between the “disc lake” and experimental design?
11. Page 8, line 11-13, IBR calculation need to provide some details. Not all physical parameters and biomarkers were used?
12. Page 11, line 37, an explanation of data analysis should be provided for “correlation coefficient square R² 0.9666”.
13. Authors used full text “Figure” in text, while used abbreviation “Fig.” in attached documents. Please check and keep the same description in the whole paper whatever in the diagram or in the text.
14. Fig. 4 was not clearly presented, Please check and revise.
15. Taking higher frequent used “High Water Level or High Water Year” for consideration, authors should mention the abbreviations appeared at the first time in text and then used abbreviation in the later description. Please Check and revise.
16. Tables 1, please supply an explanation in contents why this experimentation only simulates 6 months, which is little different in description as mentioned in introduction.
17. The language used in this paper could be further improved by native English speakers or professional English editing institutes.

Author's Response to Decision Letter for (RSOS-180729.R0)

See Appendix A.

Decision letter (RSOS-180729.R1)

18-Oct-2018

Dear Dr Ji:

On behalf of the Editors, I am pleased to inform you that your Manuscript RSOS-180729.R1 entitled "Integrated biomarker responses of the submerged macrophyte *Vallisneria spiralis* via hydrological processes from Lake Poyang, China" has been accepted for publication in Royal Society Open Science subject to minor revision in accordance with the referee suggestions. Please find the referees' comments at the end of this email.

The reviewers and Subject Editor have recommended publication, but also suggest some minor revisions to your manuscript. Therefore, I invite you to respond to the comments and revise your manuscript.

- Ethics statement

- Data accessibility

If you wish to submit your supporting data or code to Dryad (<http://datadryad.org/>), or modify your current submission to dryad, please use the following link:
<http://datadryad.org/submit?journalID=RSOS&manu=RSOS-180729.R1>

- Competing interests

- Authors' contributions

All submissions, other than those with a single author, must include an Authors' Contributions section which individually lists the specific contribution of each author. The list of Authors

should meet all of the following criteria; 1) substantial contributions to conception and design, or acquisition of data, or analysis and interpretation of data; 2) drafting the article or revising it critically for important intellectual content; and 3) final approval of the version to be published.

- Acknowledgements

- Funding statement

Because the schedule for publication is very tight, it is a condition of publication that you submit the revised version of your manuscript before 27-Oct-2018. Please note that the revision deadline will expire at 00.00am on this date. If you do not think you will be able to meet this date please let me know immediately.

- 1) A text file of the manuscript (tex, txt, rtf, docx or doc), references, tables (including captions) and figure captions. Do not upload a PDF as your "Main Document".
- 2) A separate electronic file of each figure (EPS or print-quality PDF preferred (either format should be produced directly from original creation package), or original software format)
- 3) Included a 100 word media summary of your paper when requested at submission. Please ensure you have entered correct contact details (email, institution and telephone) in your user account

- 4) Included the raw data to support the claims made in your paper. You can either include your data as electronic supplementary material or upload to a repository and include the relevant doi within your manuscript
- 5) All supplementary materials accompanying an accepted article will be treated as in their final form. Note that the Royal Society will neither edit nor typeset supplementary material and it will be hosted as provided. Please ensure that the supplementary material includes the paper details where possible (authors, article title, journal name).

Please note that Royal Society Open Science charge article processing charges for all new submissions that are accepted for publication. Charges will also apply to papers transferred to Royal Society Open Science from other Royal Society Publishing journals, as well as papers submitted as part of our collaboration with the Royal Society of Chemistry (<http://rsos.royalsocietypublishing.org/chemistry>). If your manuscript is newly submitted and subsequently accepted for publication, you will be asked to pay the article processing charge, unless you request a waiver and this is approved by Royal Society Publishing. You can find out more about the charges at <http://rsos.royalsocietypublishing.org/page/charges>. Should you have any queries, please contact openscience@royalsociety.org.

on behalf of Dr Punidan Jeyasingh (Associate Editor) and Prof. Jon Blundy (Subject Editor)
openscience@royalsociety.org

Associate Editor Comments to Author (Dr Punidan Jeyasingh):

The authors have done a great job of incorporating comments from reviewers. Consequently, this version is much improved. I am happy with it, and would like to see it published. Nevertheless, I urge the authors to give the grammar a bit more attention. For example, the first "sentence" of the abstract is 5 sentences long! The newly added text throughout the manuscript is important. It would help a lot if it is communicated more clearly. As such, I am recommending to accept this manuscript, pending minor revision.

Author's Response to Decision Letter for (RSOS-180729.R1)

See Appendix B.

Decision letter (RSOS-180729.R2)

30-Oct-2018

Dear Dr Ji,

I am pleased to inform you that your manuscript entitled "Integrated biomarker responses of the submerged macrophyte *Vallisneria spiralis* via hydrological processes from Lake Poyang, China" is now accepted for publication in Royal Society Open Science.

on behalf of Dr Punidan Jeyasingh (Associate Editor) and Prof. Jon Blundy (Subject Editor)
openscience@royalsociety.org

Appendix A

Dear Editors and Reviewers:

Sincere thanks for the editor and reviewer's comments concerning our manuscript entitled "Integrated Biomarker Responses of submerged macrophyte *Vallisneria spiralis* with the hydrological process from Lake Poyang, China" (MS Reference No: RSOS-180729). Those comments are all valuable and very helpful for revising and improving our paper, as well as the important guiding significance to our researches. All authors have studied comments carefully and have made correction according the Guidelines for authors, abstract, introduction, method, results, discussion, conclusion and figures, which we hope meet with approval. Revised portion are marked in red in the paper. The main corrections in the paper and the responds to the reviewer's comments are as flowing:

Responds to the reviewer 1's comments:

(1) Response to General comments

1) While the introduction does a good job of setting up the importance of the study, the authors could do a better job of explaining what their predictions or hypotheses were for the experiment.

Thanks for reviewer's suggestion, we reorganized the introduction part and put forward the hypothesis of this paper in time after introduction of the current situation of wetland hydrological situation of Poyang Lake and the ecological value of the representative wetland vegetation. As modified in the last two paragraphs and other parts of the introduction, the paragraph structure and parts of the content were also supplemented and revised.

Once again, thanks for your suggestion, which do make the Introduction Part more reasonable and readable.

2) The methods need much more detailed information. Were the glass jars (tanks?) outside, in a greenhouse, or indoors? Was water temperature measured? Where was the water that was in the tanks from? At what frequency did the authors add nutrients to the water? On what plant material did the authors conduct the biomarker assays? Were plants destructively sampled to determine biomass, or were they weighed while still wet? Etc...

The authors should write the methods so that a reader would have enough information to exactly replicate the study.

We have made correction in the Methods Part according to the Reviewer's comments and listed some revision as following.

2.1 Experimental facilities and water exchange

After two weeks preculture and acclimation in laboratory with dechlorinated municipal water adding Hoagland Nutrient Solution with proportion of 1:10 (Beijing Kolaibo Technology Ltd. Co.), well growth plants were rinsed with tap water and distilled water, dried by filter paper. Uniform height samples with 1.19 ± 0.12 gram (g) weigh and 10 ± 0.45 centimeter (cm) height were chose as experimental materials and planted in sandy loam soil from April to September, 2016. In order to simulate natural conditions, the whole experiment was carried out outdoor in the field of YIFU experimental building of Nanchang Institute of Technology (Nanchang, China). There was no building blocking within 100 meters of the site, and an irrigation canal flow through nearby brought from Fuhe river. As showed in Figure 2, the designed plastic bucket with 10 ± 0.5 cm thick sandy loam soil at the bottom was placed in a glass tank with length 50 cm \times width 50 cm \times height 100 cm. The 12 strains of precultured *V. spiralis* were moved from laboratory into this equipment. During the whole experiment, the water in tank was half replaced to maintain water quality and transparency with dechlorinated municipal water adding Hoagland Nutrient Solution with proportion of 1:10 (Beijing Kolaibo Technology Ltd. Co.) every week.

2.2 Definition of Disc Lake

In order to reflect the hydrological characteristics of Disc lakes where is usually referred to the lower area separated from main water and formed as an individual internal lake in dry season from National Nature Reserve in Poyang Lake (Wu Cheng), the maximum water level in high water level group (HWL), minimum water level in low water level group (LWL) and the average monthly water level in normal water level group (NWL) were selected for laboratory simulation.

2.3 Morphology and biomarker analysis

At the end of every month, three plants were taken in each parallel group and rinsed with tap water and distilled water, dried by filter paper and transported to the lab for physical measure. The morphological parameters (plant height and biomass) of *V. spiralis* were then measured that plant

height was measured by measuring scale and the biomass was determined by weighing method. The average height increment of vegetation is defined as the difference of plant height between two adjacent months, and calculated from subtracting the average height of plant in the next month by the average height in the previous month. After that, leaves were cut separately into small pieces to a size less than 1 mm by stainless steel scissors, and then separately packed into polyethylene ziplock bag, marked and numbered and then stored in the refrigerator at -40°C for biomarker assays.

Weighing 0.5 g of leaves, samples were fully ground within 9 volumes of cold buffer (0.15 M KCl, 0.1 M Tris - HCl, pH 7.4) and centrifuged for 25 min ($12,000\times g$) at 4°C . Supernatants were used as the extract for the content of protein and enzymatic activity determination. Reagent Kits provided by Nanjing Jiancheng Biological Technology co. Ltd were used for soluble protein contents, enzyme activities analysis, including superoxide dismutase (SOD), peroxidase (POD), catalase (CAT), and lipid degradation product (thiobarbituric acid, TBARS).

3) There is no explanation of which statistical tests the authors used beyond generally stating that Excel was used. It is very important that the authors choose the appropriate analysis for tests among groups, especially if the data are not normally distributed.

Considering the reviewer's suggestion, the detail of statistical tests was added and listed as showing below. Once again, thanks for your suggestion.

In the present study, the growth of plant height, aboveground and underground biomass, SOD, POD and CAT activities are combined into a IBR comprehensive index after standardization. By this method, the comparison between different experimental groups can be accessed and the visualization results will be more acceptable. The data was counted by mean value \pm standard deviation SD. The differences between the groups were tested using the LSD method in one-way ANOVA by IBM SPSS 22.0, in which significant difference was indicated with $p < 0.05$ and highly significant difference with $p < 0.01$. This study uses Microsoft Excel 2010 software to do statistics and calculation of data. Adobe Photoshop CS6 and Corel DRAW are taken for Image optimization.

4) I would also recommend that the authors pay particular attention to improving the grammar in their manuscript. Notably, much of the manuscript was written in the future tense. Given that this experiment was already conducted, the authors should never use the future tense and instead use the past tense (or present tense where appropriate).

Thanks for reviewer's suggestion, the manuscript has been polished by Elsevier for language problem. The certificated document and the revised manuscript offered by Elsevier have also be presented at the same time.

(2)Response to specific comments

ABSTRACT

Line 2: Add ",China" after Lake Poyang

We have made correction according to the Reviewer's comments.

Line 16-17: Do not use acronyms in abstracts without defining them

We have made correction according to the Reviewer's comments.

End: Try to add a general sentence that brings all of the results (including biomass, antioxidant enzyme activities, etc.) together into a main final conclusion of the study -- Something like "We found that high flooding levels had the strongest negative effect on growth and enzyme activity of *Vallisneria spiralis*"

Considering the reviewer's suggestion, a general sentence has been added into this part for complete and clear representation.

INTRODUCTION

Page 3, Line 29: What do the authors mean by "As the energy source of photosynthesis, rhythmic changes in wetland water level can effect..."? Do they mean to say that water is affecting the light levels? The way the sentence is written as of now it sounds like the authors are saying that water is the energy source

Sorry for this grammar error, we have made correction according to the Reviewer's comments.

Page 4, Line 22: It would be helpful for there to be a map of where the lake is in China, maybe in the methods?

Considering the reviewer's suggestion, the basic elements, including China, Jiangxi Province, Lake Poyang, Ganjiang River and several Disc Lakes, were added and illustrated in Figure 1 as showing below.

Page 5, Line 18: Start a new paragraph at “In order to fully...”

We have made correction according to the Reviewer’s comments.

METHODS

Page 5, Line 54: Does this weight include roots or the leaves or the whole plant? Is it a wet weight? If so, how did you control for differences in the amount of water on and in the plant?

As mentioned above, we have made correction according to the Reviewer’s comments.

Page 6, Line 3: Do not use the future tense, here or nearly anywhere else

We have made correction according to the Reviewer’s comments.

Section 2.3: This section need the most work. The explanation of the experimental setup is confusing and needs many more details related to how the experiment was run for 6 months.

As mentioned above, we have made correction according to the Reviewer’s comments.

What is a disc lake?

We have added the definition in the Method Part according to the Reviewer’s comments.

Page 7, Line 35: Remove “accurately”. Every measurement has error associated with it

We have made correction according to the Reviewer’s comments.

Page 7, Line 43: Redefine enzyme abbreviations

We have made correction according to the Reviewer's comments.

Page 7, Line 45: Provide citations for the hydroxylamine method

We have made correction according to the Reviewer's comments.

What instrument were the activities measured on?

We have made correction according to the Reviewer's comments.

Section 2.6 – This is another section that needs a lot of work. Which statistical analyses were used to test what? Describe all statistical analyses in this section.

As mentioned above, we have made correction according to the Reviewer's comments.

RESULTS

Section 3.1

What does klx stand for?

We have made revision in Result Part according to the Reviewer's comments.

How did you test for a significant difference? Please report the statistical method and the P-value

As mentioned above, we have made correction in the Method Part and added P value according to the Reviewer's comments.

I am confused by the metric plant height increment. This was not explained in the methods section

We have added the definition in the Method Part according to the Reviewer's comments.

Biomass should be normalized to area. Is this biomass within the jars? If so, were the plants destructively sampled? This also needs more explanation in the methods

We have added explanation in the Method Part according to the Reviewer's comments.

Section 3.2

The sentence "IBR values reflect the comprehension responses to the stress of anoxia condition" should be moved to the data analysis section (and the grammar should be checked). Instead, since this is the results, the authors should describe the differences among treatments in these star plots

Considering the reviewer's suggestion, we have moved the sentence "IBR values reflect the comprehension responses to the stress of anoxia condition" to the data analysis section and made grammar revision. Meanwhile, contents of the differences among treatments were also supplied

in this part.

How did the authors assess significant inhibition?

As mentioned above, we have made correction in the Method Part and added P value according to the Reviewer's comments.

Section 3.3

Much of this section should be moved to the methods because it is simply describing how IBR was used, and not what the results were.

Considering the reviewer's suggestion, much of this section has been moved to the methods.

DISCUSSION

How do you disentangle the impact of light intensity versus anoxic conditions?

Thanks for your kindly question. In fact, our research group in Nanchang Institute of Technology has been keep deep cooperation with Jiangxi provincial bureau of hydrology which is responsible for water quality monitoring of Lake Poyang. Based on Monitoring Bulletin, concentration of dissolved oxygen (DO) usually ranges from 7.05~8.66 mg/L within 5 meters and its saturation rate ranges from 78.4~99.9 % at the permanent observation stations in Lake Poyang. So, DO was not considered as a control factor in this study and not be observed. Nevertheless, thanks to the comments of reviewer, we will try our best to improve the experimental proposal at the next stage of experiments.

Page 11, Line 37: This correlation is not shown in Figures 2A and 2B, but it is good to see some reported regression values. What is the p-value of this relationship? It clearly is significant if it explained so much of the variance. Considering the Reviewer's suggestion, we have supplied corresponding contents in this part and reorganized the paragraph structure as listed following: There was a significant negative correlation with correlation coefficient square R^2 0.9666 between average light intensity under water and average height of vegetation growth performed with a polynomial regression analysis using Data from April to July ($P < 0.001$).

Were the raw data or was the median value used for the regression analysis?

As mentioned above, the average light intensity under water and average height of vegetation growth were used in this part to perform regression analysis.

Do not use the future tense. You are discussing work that has already been

done.

Sorry for this grammar error, we have made correction according to the Reviewer's comments.

Page 11, Line 54: "above ground" and "below ground" should both be one word (e.g. "aboveground")

Sorry for this grammar error, we have made correction according to the Reviewer's comments.

Page 13, Line 18: This would be a good sentence to incorporate into the end of the abstract because it does a good job of summarizing all of the results together

Considering the Reviewer's suggestion, we have added a conclusion sentence in abstract to make a final conclusion of all the results together.

Page 14, Line 22: Define NP to nanoparticles

Sorry for this grammar error, we have made correction according to the Reviewer's comments.

Page 14, Line 27: How did you assess "Well agreement"?

Thanks for reviewer's suggestion. Firstly, IBR values of the treated group were compared with that of the control group and sorted based on their deviation degree. After that, qualitative analysis was made by numerous single indicator of the experimental group to determine the adaptive effect of IBR value.

FIGURES

Figure 1. Needs to have a scale and much more information about the device, especially given that the water fluctuated.

Considering the reviewer's suggestion, the figure has been revised by adding scale and more detail.

Figure 2.

What is the random "5.29" doing in plot 2A?

Sorry for this grammar error, we have made correction according to the Reviewer's comments.

The standard deviation bars are remarkably consistent from month to month. Are the authors certain that they calculated a separate standard deviation for each month?

Thank you very much for your comments and we are so sorry for this error. After inspection, we found that the standard deviation option in EXCEL was

checked automatically when plotting, which made the standard deviation values showed in figure with same values. We have been carefully checked and revised this error. Thanks again for your comments.

How did the authors test for significant differences?

As mentioned above, we have made correction in the Method Part and added P value according to the Reviewer's comments.

Figure 3

How did the authors test for significant differences?

As mentioned above, we have made correction in the Method Part and added P value according to the Reviewer's comments.

Figure 4

Make sure that text from separate plots does not overlap in the final figure "CK" in the plots should be changed to "CG"

Thank you for your suggestions. This figure has been carefully plotted again to make sure every elements clear illustration and no overlap. Furthermore, "CK" was revised as "CG".

Figure 5.

- The legend only needs to be shown once for all four panels, preferably to the right of all four plots.

- The scale needs to be the same magnitude among all four panels (e.g. Illumination intensity needs to go from 0 to 25 on all 4 plots)

- This is a really interesting figure!

Thank you for your suggestions. This figure has been modified with legend shown once on the first panel to the right and the uniform magnitude among all four panels.

Table 1.

- "may" need to be capitalized to "May"

Sorry for this grammar error, we have made correction according to the Reviewer's comments.

Responds to the reviewer 2's comments:

1. The paper's title could be revised as "Integrated Biomarker Responses of submerged macrophyte *Vallisneria spiralis* for the hydrological process of Lake Poyang, China".

2. Page 4, line 22, change "square kilometer" to km².

We have made correction according to the Reviewer's comments.

3. Page 4, line 48, change "Poyang lake" to "Poyang Lake".

We have made correction according to the Reviewer's comments.

4. Page 5, line 2, change "biochemical of *Vallisneria spiralis*" to "biochemical indices of submerged plants".

Sorry for this grammar error, we have made correction according to the Reviewer's comments.

5. "*Vallisneria spiralis*" can be abbreviated as "*V. spiralis*" after the first use.

Considering the reviewer's suggestion, "*Vallisneria spiralis*" was used at the first time and be abbreviated as "*V. spiralis*" after that.

6. Page 6, line 11, Tan[22] should be Tan et al. [22] . Same errors in Page 12 lines 9 and 11

Sorry for this grammar error, we have made correction according to the Reviewer's comments.

7. Page 6, line 41, change 0.5cm to 0.5 cm. Pay more attention the essential space in the full text.

Sorry for this grammar error, we have made correction according to the Reviewer's comments.

8. Page 7, line 7, change "The water depth" to "the water depth".

We have made correction according to the Reviewer's comments.

9. Page 7, line 43-45, revise this sentence "enzyme activities analysis, including SOD, POD, CAT and TBARS". TBARS is not enzyme.

We have made correction according to the Reviewer's comments.

10. Page 6, line 47, Materials and Methods, authors should give an explanation what is "disc lake" and what is the relationship between the "disc lake" and experimental design?

The definition of "disc lake" has been added in this part and listed as following:

In order to reflect the hydrological characteristics of Disc lakes where is usually referred to the lower area separated from main water and formed as an individual internal lake in dry season from National Nature Reserve in Poyang Lake (Wu Cheng), the maximum water level in high water level group (HWL), minimum water level in low water level group (LWL) and the average

monthly water level in normal water level group (NWL) were selected for laboratory simulation.

11. Page 8, line 11-13, IBR calculation need to provide some details. Not all physical parameters and biomarkers were used?

Considering the reviewer's suggestion, more details has been supplied in this part as listed following: As presented in Figure 5(A)-(F), plants morphological characteristics and antioxidase stress parameters of phases I and II of *V. spiralis* were chosen to provide integrated evaluation for different hydrological progress.

12. Page 11, line 37, an explanation of data analysis should be provided for "correlation coefficient square R2 0.9666".

Considering the Reviewer's suggestion, we have supplied corresponding contents in this part and reorganized the paragraph structure as listed following: There was a significant negative correlation with correlation coefficient square R2 0.9666 between average light intensity under water and average height of vegetation growth performed with a polynomial regression analysis using Data from April to July ($P < 0.001$).

13. Authors used full text "Figure" in text, while used abbreviation "Fig." in attached documents. Please check and keep the same description in the whole paper whatever in the diagram or in the text.

Sorry for this grammar error, we have made correction according to the Reviewer's comments.

14. Fig. 4 was not clearly presented, Please check and revise.

Thank you for your suggestions. This figure has been plotted again to present clearly.

15. Taking higher frequent used "High Water Level or High Water Year" for consideration, authors should mention the abbreviations appeared at the first time in text and then used abbreviation in the later description. Please Check and revise.

We have made correction according to the Reviewer's comments.

16. Tables 1, please supply an explanation in contents why this experimentation only simulates 6 months, which is little different in description as mentioned in introduction.

Thanks for your question. In Lake Poyang, *V. spiralis* began to mature in October. Since then, the flower stalk of *V. spiralis* will be aged and

putrefaction gradually, and fruits will be gradually floated on the surface water. Based on these considerations, our experiment was only conducted until September.

17. The language used in this paper could be further improved by native English speakers or professional English editing institutes.

Thanks for reviewer's suggestion, the manuscript has been polished by Elsevier for language problem. The certificated document and the revised manuscript offered by Elsevier have also be presented at the same time.

We tried our best to improve the manuscript and made some changes in the manuscript. These changes will not influence the content and framework of the paper. And here we did not list the changes but marked in red in the revised paper. We appreciate for Editors/Reviewers' warm work earnestly, and hope that the correction will meet with approval.

Once again, thank you very much for your comments and suggestions.

Appendix B

Dear Editors:

Sincere thanks for the editor recommendation for the publication of our manuscript entitled "Integrated biomarker responses of the submerged macrophyte *Vallisneria spiralis* via hydrological processes from Lake Poyang, China" in Royal Society Open Science. All authors have studied comments carefully and have made correction according the Guidelines for ethics statement, data accessibility, competing interests, authors' contributions, acknowledgements, funding statement, and main documents grammars, which we hope meet with approval. Revised portion are marked in red in the paper. The main responds to the associate editor comments are as flowing:

Responds to the associate editor comments:

The authors have done a great job of incorporating comments from reviewers. Consequently, this version is much improved. I am happy with it, and would like to see it published. Nevertheless, I urge the authors to give the grammar a bit more attention. For example, the first "sentence" of the abstract is 5 sentences long! The newly added text throughout the manuscript is important. It would help a lot if it is communicated more clearly. As such, I am recommending to accept this manuscript, pending minor revision.

Thanks for your recommendation and comments for the revised manuscript, the manuscript has been further revised by all authors after several careful inspections marked in red in the paper. Considering the reviewer's suggestion, the beginning of Abstract has been modified as following. Once again, thanks for your suggestion.

Vallisneria spirals (*V. spiralis*), a widely distributed wetland plant, was used to reveal how the light intensity at the top of the plant, plant morphology, and antioxidant enzyme activity respond to different hydrologic conditions from Lake Poyang, China. By designing a laboratory experiment simulating historical water levels of low, normal, and high wetland plant submersion, this study aimed to elucidate the effects of different levels of flooding on growth and antioxidant enzyme activity for *V. spiralis*.

We tried our best to improve the manuscript and made some changes in the manuscript. These changes will not influence the content and framework of the paper. And here we did not list the changes but marked in red in the revised paper. We appreciate for Editors' warm work earnestly, and hope that the correction will meet with approval.

Once again, thank you very much for your comments and recommendation.